# The Efficacy of Platelet-Rich Plasma Injection Therapy in the Treatment of Patients with Achilles Tendinopathy: A Systematic Review and Meta-Analysis

**DOI:** 10.3390/jcm12030995

**Published:** 2023-01-28

**Authors:** Djandan Tadum Arthur Vithran, Wenqing Xie, Michael Opoku, Anko Elijah Essien, Miao He, Yusheng Li

**Affiliations:** 1Department of Orthopaedics, Xiangya Hospital of Central South University, Changsha 410008, China; 2National Clinical Research Center for Geriatric Disorders, Xiangya Hospital, Central South University, Changsha 410008, China

**Keywords:** Achilles tendinopathy, meta-analysis, platelet-rich plasma, randomised controlled trial, systematic literature review

## Abstract

Background: Over the past few years, many studies have been conducted to evaluate the effectiveness of platelet-rich plasma (PRP) in treating musculoskeletal conditions. However, there is controversy about its benefits for patients with Achilles tendinopathy. Objective: This study aimed to investigate whether platelet-rich plasma injections can improve outcomes in patients with Achilles tendinopathy. Methods: A comprehensive literature search was conducted in PubMed, Embase, Cochrane Library, Web of Science, China Biomedical CD-ROM, and Chinese Science and Technology Journal databases to identify randomised controlled clinical trials that compared the efficacy of PRP injection in patients with Achilles tendinopathy (AT) versus placebo, published between 1 January 1966 and 1 December 2022. Review Manager 5.4.1 software was used for the statistical analysis, and the Jadad score was used to assess the included literature. Only 8 of the 288 articles found met the inclusion criteria. Results: Our work suggests that: The PRP treatment group had a slightly higher VISA–A score than the placebo group at 6 weeks [MD = 1.92, 95% CI (−0.54, 4.38), I^2^ = 34%], at 12 weeks [MD = 0.20, 95% CI (−2.65 3.05), I^2^ = 60%], and 24 weeks [MD = 2.75, 95% CI (−2.76, 8.26), I^2^ = 87%]). However, the difference was not statistically significant. The Achilles tendon thickness was higher at 12 weeks of treatment in the PRP treatment group compared to the control group [MD = 0.34, 95% CI (−0.04, 0.71), *p* = 0.08], but the difference was not statistically significant. The VAS-improvement results showed no significant difference at 6 and 24 weeks between the two groups, respectively (MD = 6.75, 95% CI = (−6.12, 19.62), I^2^ = 69%, *p* = 0.30), and (MD = 10.46, 95% CI = (−2.44 to 23.37), I^2^ = 69%, *p* = 0.11). However, at 12 weeks of treatment, the PRP injection group showed a substantial VAS improvement compared to the control group (MD = 11.30, 95% CI = (7.33 to 15.27), I^2^ = 0%, *p* < 0.00001). The difference was statistically significant. The return to exercise rate results showed a higher return to exercise rate in the PRP treatment group than the placebo group [RR = 1.11, 95% CI (0.87, 1.42), *p* = 0.40]; the difference was not statistically significant. Conclusion: There is no proof that PRP injections can enhance patient functional and clinical outcomes for Achilles tendinopathy. Augmenting the frequency of PRP injections may boost the outcomes, and additionally, more rigorous designs and standardised clinical randomised controlled trials are needed to produce more reliable and accurate results.

## 1. Introduction

In orthopaedics outpatients, Achilles tendinopathy (AT) can be difficult for doctors since this frequent musculotendinous condition causes discomfort, impaired function, and decreased activity tolerance [1,2]. The clinic’s cautious therapies are used for this condition, which has major implications. However, these therapeutic techniques have limited curative impact, and the disease quickly relapses. [3] Long-term inappropriate or excessive activity causes the Achilles tendon to rub or overstretch beyond its healing capacity and induces inflammatory changes in the tendon and surrounding tissues. Chronic inflammation causes tendon hyaline and fatty tissue degradation. The Achilles tendon might tear due to this impact. [4] The Achilles tendon contains tendon cells, water, and fibrin collagen. It heals slower than other injured connective tissues because it has an insufficient blood supply [5]. Several surgical and non-surgical treatment options have been used to improve AT’s structural and functional prognosis [6,7]. Non-surgical treatment options include steroid injections, oral non-steroidal anti-inflammatory drugs (NSAIDs), external drug application, and low-frequency ultrasound stimulation. Steroid injections have strong anti-inflammatory and analgesic effects and are the most widely used treatment in clinical practice. However, repeated multiple injections can lead to collagen necrosis and may reduce the mechanical properties of the Achilles tendon. [8,9,10] Long-term use of NSAIDs can easily lead to gastrointestinal ulcers [11]. The treatment results are often not entirely satisfactory, leading to limitations in work capacity and reduced motor function for up to several months, ultimately affecting quality of life [12]. Therefore, there is an urgent need to find a new treatment method to accelerate tissue recovery in AT. With the development of relevant research, scholars have discovered that growth factors play a crucial role in Achilles tendon repair and have considered using platelet-rich plasma (PRP) to treat these Achilles tendon disorders.

High platelet counts and supraphysiological concentrations of platelet-derived growth factors, chemokines, and cytokines are necessary for the tissue healing and regenerative characteristics of platelet concentrate, such as PRP, platelet-rich fibrin (PRF), and concentrated growth factor (CGF) [13,14]. Preparing PRP is simple and rapid, and the quality of PRP products may be evaluated by determining the percentages of platelets, leukocytes, and growth factors present.

In recent years, there has been a dramatic increase in the clinical application of platelet-rich plasma (PRP), particularly in the fields of orthopaedics [15], sports and musculoskeletal medicine [16], aesthetic and plastic surgery [17,18,19,20,21], oral and maxillofacial surgery [10,22], and dermatology [23,24,25]. The Food and Drug Administration has approved some commercial formulation systems of PRP. By being injected intradermally or applied topically, PRP can transport biomolecules directly to the injury site. Platelet-rich plasma (PRP) injections are thought to speed healing by triggering the body’s natural tendon repair processes. Due to the lack of clinical data, the effectiveness of PRP injections for AT healing has been the subject of several clinical studies; however, more conclusive evidence of their use still needs to be provided.

Therefore, we performed a meta-analysis on PRP injections’ efficacy in treating Achilles tendinopathy to provide a more reliable basis for further guidance on clinical application.

## 2. Materials and Methods

Before beginning the investigation, we developed a prospective protocol including our goals, literature search techniques, eligibility criteria, outcome measures, and statistical analyses following the Preferred Reporting Items for Systematic Reviews and Meta-analysis (PRISMA) guidelines. PROSPERO registration number CRD42022380617.

### 2.1. Study Design

This study used meta-analysis to quantify studies on platelet-rich plasma in Achilles tendinopathy from 1966 to the present. The study design was divided into the following: (1) Setting the research plan and inclusion/exclusion criteria for the literature following the purpose of this study; (2) Searching major target databases and conducting literature searches using the search terms and search formulas that have been developed; (3) Screening eligible studies against the established inclusion and exclusion criteria; (4) Applying the Jadad scale to evaluate the quality of the literature and eliminate low-quality studies; (5) Using the data extraction form to extract the required data in detail; (6) Applying the Review Manager 5.4.1 to data entry for analysis; (7) Analyse and interpret the results and draw conclusions.

### 2.2. Data Collection

#### 2.2.1. Inclusion Criteria

The studies were included if the following criteria were met: (1) Study type: Randomised controlled trials (RCTs) published or made public between 1 January 1966 and 1 December 2022; (2) Type of patients: Patients diagnosed with Achilles tendinopathy; (3) Type of intervention: The experimental group was treated non-surgically with local PRP injection; (4) Outcome indicators: including the Victorian Institute of Sports Assessment–Achilles (VISA–A) score, change in Achilles tendon thickness, Visual Analogue Scale (VAS) for pain, patient satisfaction, return to sport, and return to exercise.

#### 2.2.2. Exclusion Criteria

Studies that have been excluded: (1) Non-randomised controlled trials, low-quality randomised controlled trials, animal and in vitro experiments; (2) Studies with follow-up less than 6 weeks after intervention; (3) Studies that did not have a control group; (4) Studies on surgical treatment or PRP in combination with other drugs; (5) Studies that did not reveal outcomes.

### 2.3. Search Methods

The search strategy was based on the Cochrane Collaboration’s suggested search strategy title, and abstract searches were performed using MeSH and Emtree words and their combinations: “platelet-rich plasma”, “Plasma, platelet-rich”, “Plasma, platelet-rich”, “plasma/platelet-rich fibrin”, “Achilles Tendon”, “Tendon, Achilles” “Tendo Calcaneus”, “Tendo Calcaneus”, “CalcanealTendons”, “Calcaneal Tendons”, “CalcanealTendons”, “tendinopathy”, “Tendinopathies”, “Tendonopathy”, “Tendinosis”, and “Tendonitis”. Medline (1966 to December 2022), PubMed (1966 to December 2022), Embase (1966 to December 2022), and the Cochrane Library (1966 to December 2022) were searched for the terms “platelet-rich plasma”, “platelet gel”, “platelet concentrates”, “PRP”, “Achilles tendinopathy”, and “Achilles tendinosis”. The period from 1966 to December 2022 was systematically searched in Embase (1966 to December 2022) and the Cochrane Library, with no language restrictions, with “platelet-rich plasma”, “platelet concentrates”, “Achilles tendon injury”, “Achilles tendinopathy”, and “Achilles tendinopathy”. “Achilles tendinopathy”, “Achilles tendinopathy”, “Achilles tendinitis”, and “chronic Achilles tendinitis” were used as Chinese search terms in the Search Wanfang Data Knowledge Service Platform (1997—December 2022), China Biomedical CD-ROM Database (1978—December 2022), China Biomedical Literature Database (1978—December 2022), and Vipshop Chinese Science and Technology Journal Database (1978—December 2022). We downloaded and read the full text of the potential literature and carefully read its references.

### 2.4. Literature Selection

At least two people worked independently to search the literature. They entered databases according to predefined search terms and search formulas. They combined the results of screening through the literature management software, eliminating duplicate publications, reading the title list to exclude irrelevant literature, downloading the full text of potentially relevant literature, reading it carefully, and deciding on the final selection of literature based on the inclusion and exclusion criteria. Furthermore, they sought third-party resolution if they reached different conclusions through arbitration, or resolved issues through discussion (Figure 1).

From the databases, 405 articles were selected, and 104 duplicate records were removed. 301 records were screened, of which 290 needed to be excluded for different reasons (100 studies included surgery, 55 studies included no control group, 53 studies included Achilles tendon rupture, 52 studies included other systemic conditions, 10 included major injuries to the ankle, and 20 studies included combined therapy.) A total of 11 articles were then assessed for eligibility. However, during our second screening, we found 3 articles excluded for other reasons: 1 article was experimental research, 1 was a retrospective case study, and 1 had no primary outcome indicator. In the end, only 8 high-quality articles were selected and included in our study.

### 2.5. Quality Evaluation of the Included Studies

The study design protocol, randomisation method concealment and blinding implementation, and the number of missed visits described in the literature were analysed and scored methodologically using a modified Jadad scale, which included four items: generation of random sequences, randomisation concealment, blinding, and withdrawal and exit, out of a total score of 7. Studies with a total score of 1–3 were considered low-quality, and those with a total score of 4–7 were considered high-quality. Those with a score of ≥ 3 were included in this study.

### 2.6. Statistical Analysis

The statistical software used for this study was Review Manager 5.4.1 (Revman 5.4), a dedicated systematic evaluation software provided by the Cochrane Collaboration that is currently recognised by evidence-based medical scholars worldwide. All eligible data from the literature were extracted according to the evaluation criteria and entered into Revman 5.4 for analysis. For count data (e.g., return to sport, the incidence of adverse events), the risk difference (RD) or the odds ratio (OR) was used to describe the data. The mean difference (MD) was used for measures (e.g., VISA–A score, VAS score, ankle mobility, etc.). Heterogeneity was tested using the I^2^ index to reflect the severity of heterogeneity; I^2^ < 31% were homogeneous across studies, I^2^ > 56% were more heterogeneous, 56% < I^2^ < 70% could not be excluded from heterogeneity, I^2^ < 50% used the fixed effects model PETO method to combine effect sizes; I^2^ > 50% used the random effects model DerSimonian–Laird method for calculation. If heterogeneity between groups was too large, meta-analysis was discarded in favour of descriptive analysis.

### 2.7. Sensitivity Analysis

The veracity and stability of the evidence largely determine the ability of evidence-based medicine to guide clinical practice. Obtaining an accurate and stable result is a crucial part of meta-analysis. In the meta-analysis process, researchers should avoid heterogeneity arising from subjective factors. The practical methodological application can reduce the generation of bias. Most of the decision-making process is clear and uncontroversial, but some decisions are subjective, arising when there are differences in the units and expressions of values included in practice, there is an inability to obtain original data, a study does not report the required information, a question is not answered according to the best statistical method, etc. Therefore, it is necessary to always consider the robustness of each step in implementing meta-analysis and whether it impacts the combined results, hence the need for sensitivity analysis. For example, it is possible to analyse whether the synthesis results have been affected by changing the inclusion and exclusion criteria, the statistical method, the choice of effect size, and the exclusion of certain literature. If there were significant changes before and after the results, this indicates significant heterogeneity in the factors that are relevant to the effect of the intervention, and conclusions should be drawn with great caution in interpreting the results.

### 2.8. Publication Bias

Error generation is unavoidable in any study, and the error caused by systematic error is called bias. In theory, bias should not occur and should be minimised in the research process. Because bias is inevitable in meta-analysis, resulting from the publication of the literature, the development of inclusion and exclusion criteria, the literature retrieval, the data extraction, and other processes, bias can cause the results of meta-analysis to deviate from the true values or even produce the opposite results, misleading clinical decisions. In a meta-analysis, publication bias has the greatest impact on results and is difficult to control. Therefore, controlling and identifying publication bias has become an important and difficult task in meta-analysis. When a study is conducted, authors are happy to participate in the submission process after a positive result is obtained, and journals are happy to accept positive results for publication. In the case of a negative result, authors are less enthusiastic about writing, the impact factor of the published journal is usually low, and the study sponsor prefers to keep the negative result unpublished. The main methods used to identify publication bias are the funnel plot, fail-safe factor, and cut-and-patch methods. In this paper, the funnel plot method was used to determine whether there was publication bias.

## 3. Result

### 3.1. Literature Screening Results

After extensive searches in major literature databases combined with the online search function of the Endnote literature management software, duplicates were removed through the software check function, and the titles and abstracts were browsed to determine whether the literature met the inclusion criteria initially. For literature that could not be definitively included, the full text was obtained and read carefully for further screening. Duplicate publications of the same study were excluded based on authorship, the number of case studies, interventions, and sites of implementation.

### 3.2. General Information on the Included Literature

The eight randomised controlled trials (526 cases in total) selected were all full-text published literature, eight in English and none in Chinese. The screening was conducted on a total of eleven publications, and after more evaluation, three were ruled out: One was a trial design protocol with no signs of how the trial turned out. One was not a real RCT; it was just a retrospective case-control study. One did not have any main outcome indicators. The final results included a total of eight publications (Figure 1). Five hundred twenty-six people with Achilles tendinopathy received PRP injections in the eight RCTs. All of them had Jadad scores of more than 3. Five studies used saline as a control group, and three used blank controls. The main features of the included studies are shown in (Table 1). All included papers were compared at the start, and there were no big differences between the two groups regarding age, weight, gender, or treatment. Figure 2, Figure 3 and Figure 4 show how the included studies were evaluated regarding the risk of bias.

### 3.3. Meta-Analysis Results

#### 3.3.1. Results of the Analysis of AT

##### VISA–A Score Description

A disease-specific questionnaire called VISA–A is used to estimate the severity of Achilles tendinopathy. The survey is designed to be completed independently. For both patients and healthcare professionals, it is simple and rapid. The eight questions in the final VISA–A questionnaire comprised the three categories of pain (1–3), function (4–6), and activity (7–8). Questions one through seven can receive a score of 10, and question eight can receive up to 30. The sum of the scores yields a score out of 100. Asymptomatic individuals receive a score of 100. Participants must only respond to portions A, B, or C of question eight. A person instantly loses at least 10, and sometimes up to 20, points if they experience pain while participating in a sport.

##### VISA–A Score Result

A total of eight papers with a total of 526 patients correctly and reasonably described the VISA–A score change values. There was statistical heterogeneity between studies (I^2^ = 76%, *p* < 0.00001), so a random effects model was used to calculate the combined statistic, which showed that the PRP treatment group (experimental group) had a higher VISA–A score than the placebo group (control group) [MD = 1.20, 95% CI (−0.94, 3.34), *p* = 0.27] (Figure 5), with no statistically significant difference. This indicates that the PRP treatment group did not significantly improve the VISA–A score. Using the Revman 5.4 software to transform the fixed and random effects models, the included studies were re-combined to calculate statistics, showing no difference between the PRP treatment group and the placebo group in the mean and 95% confidence interval of the VISA–A score. After excluding each study once individually, the combined effect sizes obtained from the new Meta-analysis were not significantly different from the total effect sizes. The above sensitivity analyses showed that the meta-analysis results were robust and reliable.

##### Changes in Achilles Tendon Thickness

A total of 98 patients in three publications correctly and reasonably described changes in Achilles tendon thickness. There was statistical heterogeneity between studies (I^2^ = 61%, *p* = 0.08), so a random effects model was used to calculate the combined statistic, which showed that 12 weeks after treatment, Achilles tendon thickness was higher in the PRP-treated group (experimental group) than in the placebo group (control group) [MD = 0.34, 95% CI (−0.04, 0.71), *p* = 0.08] (Figure 6), but the difference was not statistically significant. Using the Revman 5.4 software to transform the fixed effects model and random effects model, the included studies were re-combined to calculate statistics, showing no difference between the PRP treatment group and the placebo group in the mean and 95% confidence intervals for change in Achilles tendon thickness. After excluding each study once individually, the combined effect sizes obtained from the new meta-analysis were not significantly different from the total effect sizes. The above sensitivity analyses showed that the meta-analysis results were robust and reliable.

##### VAS Scoring

A total of 93 patients in three publications correctly and reasonably described the VAS score change for pain at the 6-,12-, and 24-week follow-ups using a 0 to 100 grading scale. Our results showed no significant difference in VAS score improvement at 6 and 24 weeks between the PRP injection group and the control group, respectively (MD = 6.75, 95% CI = (−6.12, 19.62), I^2^ = 69%, *p* = 0.30) and (MD = 10.46, 95% CI = (−2.44 to 23.37), I^2^ = 69%, *p* = 0.11). However, at 12 weeks of treatment (MD = 11.30, 95% CI = (7.33 to 15.27), I^2^ = 0%, *p* < 0.00001) (Figure 7) we observed a significant improvement in the PRP treatment group compared to the placebo group. Using the Revman 5.4 software to transform the fixed effects model and the random effects model, the included studies were re-combined to calculate statistics, showing essentially no difference between the PRP treatment group and the placebo group in terms of mean VAS scores and 95% confidence intervals. After excluding each study once individually, the combined effect sizes obtained from the new meta-analysis were not significantly changed compared to the total effect sizes. The above sensitivity analyses showed that the meta-analysis results were robust and reliable.

##### Patient Satisfaction

A total of four papers with 222 patients correctly and reasonably described improvements in patient satisfaction. There was no statistical heterogeneity between studies (I^2^ = 0%, *p* = 0.68), so a fixed effects model was used to calculate the combined statistic. The results showed that patient satisfaction was higher in the PRP-treated group (experimental group) than in the placebo group (control group) [RR = 1.07, 95% CI (0.84, 1.35), *p* = 0.58] (Figure 8), a statistically insignificant difference. This indicates that the PRP treatment group did not significantly improve patient satisfaction relative to the placebo group. Using the Revman 5.4 software to transform the fixed effects model and the random effects model, the included studies were re-combined to calculate statistics, showing no difference between the PRP treatment group and the placebo group regarding patient satisfaction, and 95% confidence intervals. The combined effect sizes obtained from the new meta-analyses were not significantly different from the total effect sizes after excluding each study individually once. The above sensitivity analyses showed that the meta-analysis results were robust and reliable.

##### Return to Sport

A total of 199 patients in four publications correctly and reasonably described the return to exercise rates. There was no statistical heterogeneity between studies (I^2^ = 0%, *p* = 0.54), so a fixed effects model was used to calculate the combined statistic, which showed that the return to exercise rate was higher in the PRP-treated group (experimental group) than in the placebo group (control group) [RR = 1.11, 95% CI (0.87, 1.42), *p* = 0.40] (Figure 9), a difference that was not statistically significant. This indicates that the PRP treatment group did not show a significantly increased rate of return to exercise relative to the placebo group. Using the Revman 5.4 software to transform the fixed and random effects models, the included studies were re-combined to calculate statistics, showing no difference in return to exercise rates and 95% confidence intervals between the PRP treatment group and the placebo group. After excluding each study once individually, the combined effect sizes obtained from the new meta-analysis were not significantly different from the total effect sizes. The above sensitivity analyses showed that the meta-analysis results were robust and reliable.

## 4. Discussion

PRP has been utilised in the clinic for a long time, and its usefulness in treating AT remains debatable. In this study, an examination of eight high-quality RCTs was conducted, and a further conclusion was drawn.

AT has several causes, and its specific process is unknown. Most studies have linked AT to overwork, improper exercise, stiff limbs, and weakness-related anatomical abnormalities. Several mechanisms produce local Achilles tendon irritation. Degenerative changes and Achilles tendon rupture may follow [34]. The Achilles tendon receives insufficient blood supply. It heals more slowly than other connective tissues. Scientists have discovered that growth factors are essential to Achilles tendon recovery and suggest using PRP to treat AT.

PRP has been shown to treat AT in several laboratories and limited clinical studies; hence, it is frequently used in clinical practice [35]. PRP relieves pain and improves patient satisfaction with tendon diseases, according to Murawski [36]. The lack of a reference group, sickness specificity, measurement, and blinding characterise such research.

Injections of PRP are widely utilised in therapeutic settings to enhance healing and regeneration. Platelet concentration in PRP is two to six times more than in whole blood [23]. Once activated, concentrated platelets can theoretically release greater than physiological levels of autologous growth factors to enhance healing and regeneration, such as in musculoskeletal treatment [24,25]. PRP can also stimulate the repair of Achilles tendon injuries and enhance their biomechanical function [37,38,39,40,41]. In tests of biochemical immune function in humans, topical treatment of PRP was found to enhance collagen I levels, decrease cell counts, increase glycosaminoglycan levels, and decrease vascular proliferation relative to controls [42]. PRP injection therapy is very recent, with just 11 high-quality RCT trials on the treatment of Achilles tendon disease [26,27,28,29,30,43,44,45,46,47], and no consensus has been reached about their findings.

Therefore, we conducted a meta-analysis to aggregate these disparities and provide recommendations for using PRP in treating AT. In this meta-analysis, we found moderate evidence that the PRP injection group results were not superior to the placebo group regarding patient outcomes such as VISA–A scores, patient satisfaction, return to sports rates, and VAS Scores of patients at 6 and 24 weeks. Although the Achilles tendon thickness was higher in the PRP-treated group at 12 weeks of treatment, the difference was not statistically significant; on the other hand, the VAS score of patients in the PRP group shows an improvement compared to that of the placebo group at 12 weeks of treatment. According to these results, our study did not support using PRP injections as a non-surgical therapy for Achilles tendinopathy.

Only high-quality RCTs were included in this meta-analysis to examine the simultaneous impact of PRP injection on Achilles tendinopathy. Most of the included randomised controlled trials demonstrated allocation concealment, participant blinding, and outcome assessment information. A large number of randomised controlled trials would have improved the reliability of our findings. In addition, we analysed several subgroups depending on Achilles tendon lesions and the duration of follow-up.

A study by De Jonge examined the effects of PRP and placebo injections alone or with eccentric training on pain and function in tendinitis patients. According to De Jonge, PRP injection or placebo did not significantly enhance pain relief or function in tendinitis patients. PRP has been used to treat chronic tendinopathy; therefore, this study has therapeutic implications. In a double-blind RCT, De Vos administered PRP to 54 chronic AT patients; PRP injection did not improve the ultrasonic echo structure and neovascularisation score of the Achilles tendon lesion. These studies do not support the therapeutic usage of PRP. Both studies found no therapeutic benefit of PRP injection over placebo. According to Krogh et al., PRP injection did not improve AT [29]. However, patients were blindfolded, making it difficult to estimate PRP’s late effects.

Verrall et al. [48] observed that mobility during AT treatment was equivalent to stopping exercise for 6 weeks. Rest may improve prognosis. Van der Plas et al. [49] found that in 46 patients (58 AT), the VISA–A score significantly increased from 49.2 at baseline to 83.6 after 5 years (*p* < 0.001) of eccentric exercise. At follow-up, 39.7% reported pain relief, and the sagittal Achilles tendon thickness decreased from 8.05 to 7.50 mm. Eccentric exercise eases chronic tendinitis and accelerates tendon remodelling and tissue recovery. [50]

Patient-derived PRP is administered after in vitro centrifugation. Thus, PRP does not cause immunological rejection or disease. This is confirmed in other illnesses besides AT. PRP has mild, temporary side effects [51]. This study showed no adverse reactions after tendon PRP injection.

Our study presents several limitations; therefore, its conclusions should be interpreted cautiously. 

Foremost, each PRP synthesis procedure provides products with different biological functions; thus, there are no universal standards for producing and using PRP in basic or clinical research. While most studies have used equal frequencies of PRP injections in intervention groups, PRP injection quality and amount may differ, resulting in clinical heterogeneity and ambiguity. The Food and Drug Administration has approved various commercial PRP formulation designs, but worldwide formulation standards and quality evaluation have not yet been established. In the included studies, blood collection volume, centrifugation speed/time, and platelet activation techniques varied, influencing PRP preparation yield, purity, viability, and platelet activation status, as well as quality and quantity. A study by Keene et al. [45] detailed the exact amounts of platelets, leukocytes, and representative growth factors in their PRP injections. Platelet-derived growth platelet concentrations in PRP injections were recorded in just two studies [47,52]. Although most approved studies injected 4 millilitres of platelet-rich plasma fluid in randomised controlled trials, changes in platelet, leukocyte, and growth factor concentrations can affect AT’s clinical and functional outcome. Rossi et al. [53] summarise other major PRP classification systems [54,55,56,57,58,59]. Thus, none of those mentioned classification schemes was standardised. In 2018, the ISTH Scientific and Standardisation Committee (SSC) Working Group of the Platelet Physiology Section proposed and published a formal categorisation system and consensus technique (RAND method) to regulate platelet product utilisation. These 45 recommendations contain an overview, platelet preparation, clinical trial design, and platelet utilisation in various therapeutic settings [60]. However, in actual practice, this RAND PRP technique might be time-consuming and costly for clinicians.

Second, meta-analysis should consider AT participants’ baseline demographic and clinical data (e.g., age, gender, time of AT injury). PRP injections in different countries use different clinical techniques. Clinical variability allows subgroup analysis; the absence of relevant knowledge on these potentially confounding variables prevented this investigation.

Thirdly, our meta-analysis may not apply to all AT patients; previous RCTs utilised a single injection of PRP solution, but Boesen et al. [30] administered four injections two weeks apart, which may explain the divergent findings. Platelets in PRP release most growth factors from the alpha-granules shortly after activation, and some can be inactivated in situ. Therefore, a single PRP injection is insufficient to induce tendon regeneration and repair. According to Abate et al., patients with persistent tendinopathy with several PRP injections showed improved function and discomfort [61]. PRP may help repair tendinitis, although its mechanism is unknown. In this case, we can assume that multiple PRP injections can enhance AT’s healing process. Thus, future randomised controlled trials will investigate the prognostic advantages of multiple PRP injections.

## 5. Unanswered Questions and Prospects

The clinical usage of PRP has increased due to technological developments in PRP devices and preparation. PRP biologic preparation techniques and biological characteristics are yet unclear. PRP indications and uses have yet to be fully explored. PRP was commercially accessible as an autologous blood-derived product until recently, allowing clinicians to apply platelet growth factor technology in certain pathologies and disorders. PRP success was initially exclusively based on platelet concentrations above whole blood levels. Fortunately, academics now understand PRP better. This study acknowledges that PRP preparation procedures still lack standardisation and categorisation; hence, there is no consensus on PRP biologics. More research supports platelet dosage amounts that boost angiogenesis. This involves studying platelet processes, PRP effects on leukocytes, MSCs, and intercellular interactions. Leukocytes in PRP preparations reveal harmful or helpful effects. Platelets and the innate and adaptive immune systems have been linked. Rigorous and well-documented clinical research must determine PRP’s potential and therapeutic efficacy. As treatment effects are unpredictable, future studies should include substantial sample numbers and robust design methods. Achilles tendon pathology diagnostic and imaging criteria must be universal to avoid patient heterogeneity. PRP should be examined by age and sex. PRP should be tested for minimally invasive tendon suturing. We must determine if ultrasound-guided PRP injection improves tendon repair. PRP manufacturing needs global standardisation.

## 6. Conclusions

Our study shows moderate evidence that RPP injection did not significantly improve VISA–A scores, patient satisfaction, or return to sports rates, and VAS improvement results showed no significant difference at 6 and 24 weeks between the two groups. Nevertheless, the Achilles tendon thickness was greater in the PRP-treated group than in the placebo group at 12 weeks of treatment; the difference was not statistically significant. On the other hand, at 12 weeks of treatment, the PRP injection group showed a substantial VAS improvement compared to the control group, and the difference was statistically significant. Our findings did not support the utility of PRP injection for non-surgically treated Achilles tendinopathy. It is evident that our study did not reach a consensus with previous research; therefore, we encourage researchers and orthopaedic physicians to keep an open eye on the topics. More rigorous designs and standardised methods are needed to produce more reliable and accurate results.

## Figures and Tables

**Figure 1 jcm-12-00995-f001:**
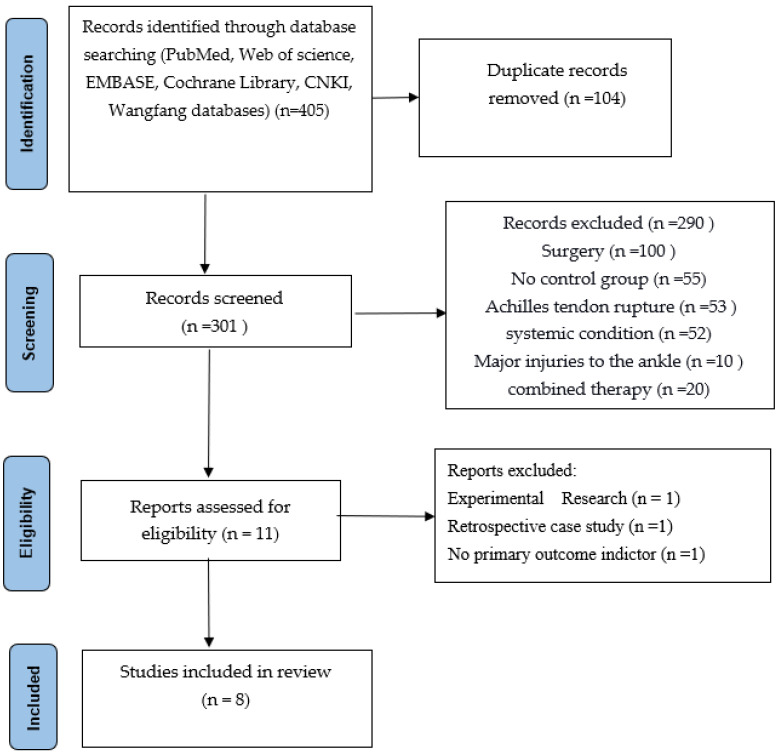
Literature screening flow chart.

**Figure 2 jcm-12-00995-f002:**
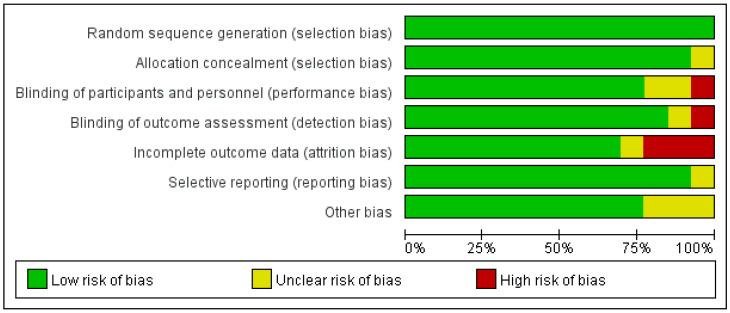
Proportional risk of bias graph: judgement of the percentage of bias items arising from all included studies.

**Figure 3 jcm-12-00995-f003:**
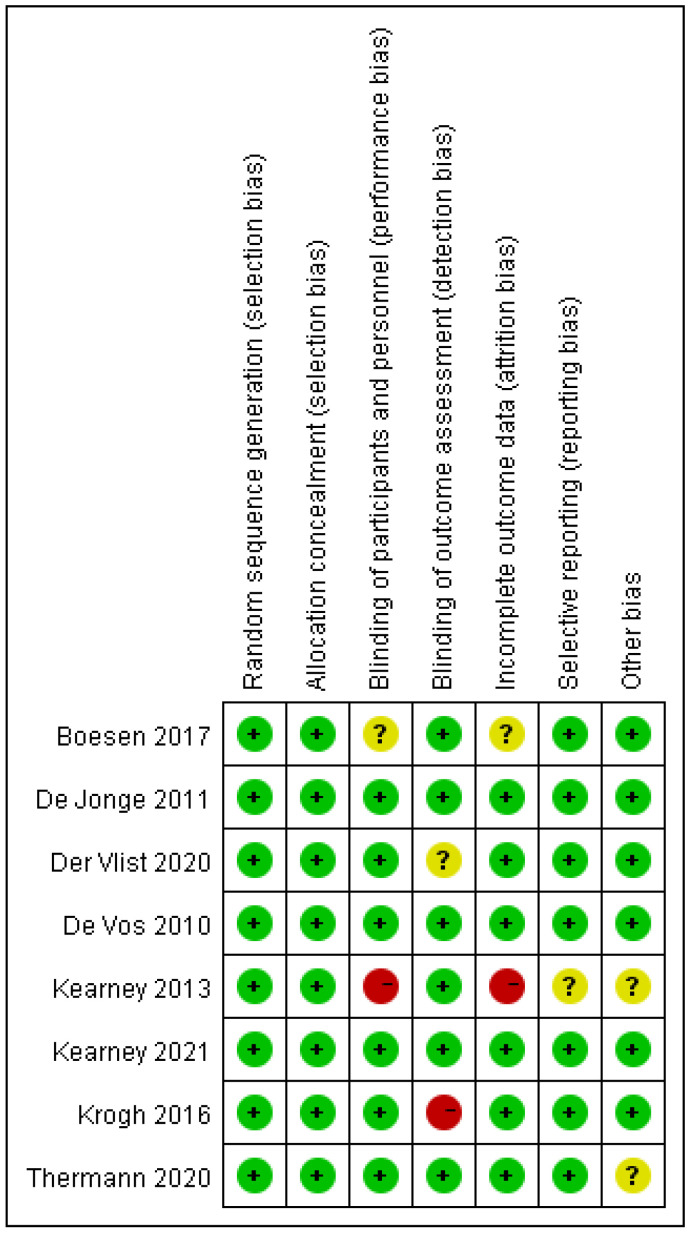
Risk of bias summary diagram: assessment of specific items of bias arising from all included studies.

**Figure 4 jcm-12-00995-f004:**
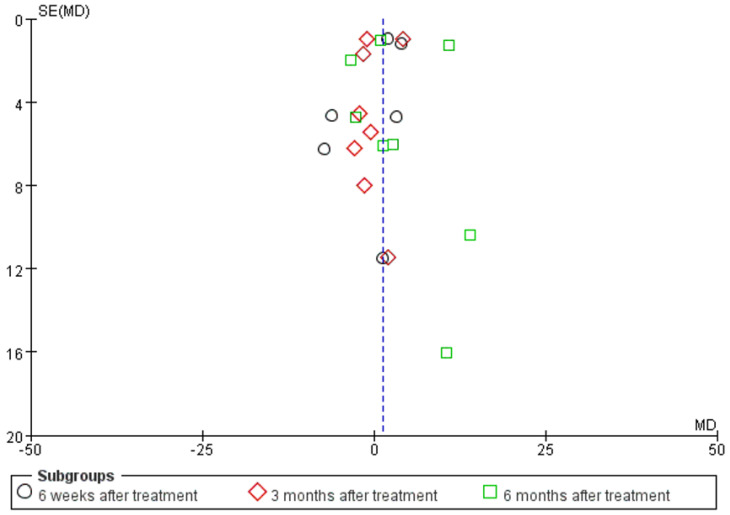
Funnel plot based on the Victoria College Ankle Function Scale scores from eight studies of Achilles tendinopathy.

**Figure 5 jcm-12-00995-f005:**
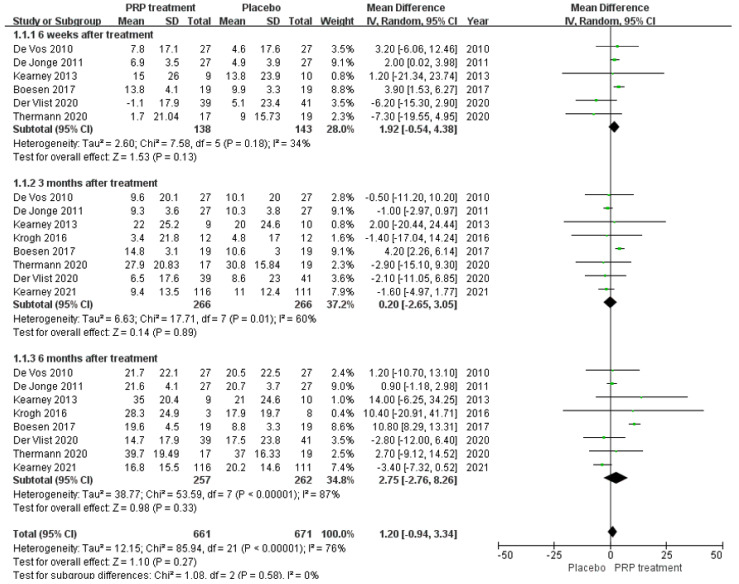
Forest plot of change values for the Victorian College Ankle Function Scale scores.

**Figure 6 jcm-12-00995-f006:**
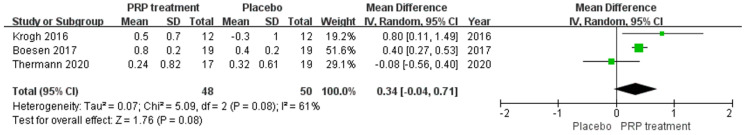
Forest plot of Achilles tendon thickness variation.

**Figure 7 jcm-12-00995-f007:**
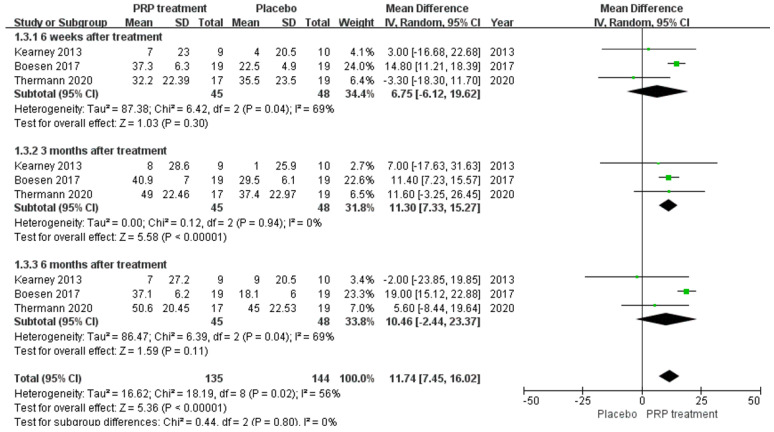
Forest plot of the change in visual analogue score.

**Figure 8 jcm-12-00995-f008:**
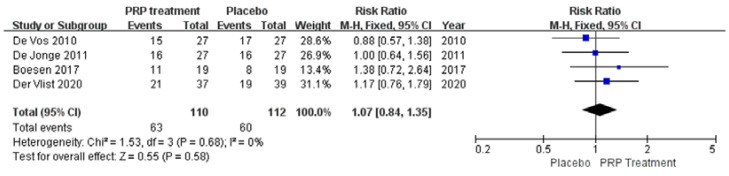
Forest plot of changes in patient satisfaction.

**Figure 9 jcm-12-00995-f009:**
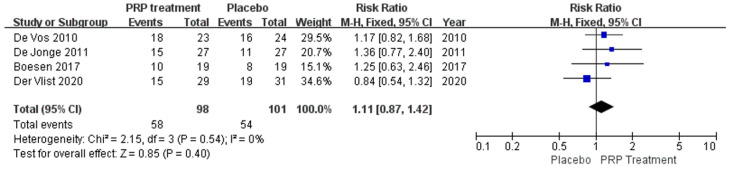
Forest plot of patient return to exercise rate.

**Table 1 jcm-12-00995-t001:** Characteristics of the included studies.

Study	Year	Control	SD	Sample Size (I/C)	Achilles Tendon Lesion	Location	PRP Injection Frequency/Interval/Dose (mL)	Follow-Up (wks/mos)
De Vos [26]	2010	Saline	RCT	27/27	C-AT (>2 mos)	Netherlands	Once/-/4	6. 12. 24 weeks
DeJonge [27]	2011	Saline	RCT	27/27	C-AT (>2 mos)	Netherlands	Once/-/4	6. 12. 24. 48 weeks
Kearney [28]	2013	Blank	RCT	9/10	C-AT (>8 mos)	UK	Once/-/3 to 5	6 wks. 3 mos, 6 mos
Krogh [29]	2016	Saline	RCT	12/12	C-AT (mean 33 mos)	Denmark	Once/-/6	3. 6. 12 mos
Boesen [30]	2017	Saline	RCT	19/19	C-AT (>3 mos)	Denmark	4 times/2-wks/4	6. 12. 24 weeks
Van der Vlist et al. [31]	2020	Saline	RCT	39/41	C-AT (>6 mos)	Netherlands	Once/-/NR	2. 6. 12. 24 weeks
Thermann [32]	2020	Blank	RCT	17/19	C-AT (>6 mos)	Italy	Once/-/NR	6 wks. 3. 6. 12 mos
Kearney [33]	2021	Blank	RCT	121/119	C-AT (>3 mos)	UK	Once/-/3	2 wks. 3. 6 mos

PRP: platelet-rich plasma; C-AT: chronic Achilles tendinopathy; wks: weeks; mos: months; RCT: randomised controlled trial; PRP acquiring ratio = blood volume (mL): PRP acquiring ratio (mL); NR: not reported, SD = Study Design.

## Data Availability

The data that support this study’s findings are available in this article.

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
