# Peer review of "The Efficacy of Platelet-Rich Plasma Injection Therapy in the Treatment of Patients with Achilles Tendinopathy: A Systematic Review and Meta-Analysis"

_jcm, 2023, doi:10.3390/jcm12030995_

Round 1

Reviewer 1 Report

  1. The manuscript requires English language proofreading and editing.

  2. The manuscript requires analysis by a statistical editor.

  3. Please, give a quick description of the VISA-A questionnaire and VAS scale. What are the treatment aims (higher VISA-A, lower VAS, lower Achilles thickness after treatment, greater difference between Achilles tendon thickness pre- and pos- treatment)?

  4. Please, rewrite the Objective for a clear meaning, eg.: The study aimed to determine the efficacy of PRP in the treatment of Achilles tendonitis.

  5. How was the Achilles tendon thickness determined in the analysed studies (ultrasound, MRI)?

  6. The first paragraph of the Results section of the Abstract belongs to the Methods section and should be transferred there.

  7. What exactly was taken into consideration in the Achilles tendon thickness analysis? Was is the difference between the tendon thickness before and after PRP treatment? This is not stated clearly at any point in the text.

  8. The description of the analysis of the VAS score results is unclear. Please, rewrite for a clear meaning both in the Abstract and in the Results section of the manuscript's main body. How would you explain the lack of significant difference in VAS score results after 6 weeks and 6 months but the presence of significant difference after 3 months? Could it be due to bias? These results do not justify the statement contained in the Conclusions section of the Abstract that PRP significantly improved VAS scores.

  9. Introduction: „periorbital tissue” is incorrect in this context as it refers to the eye. Please, correct.

  10. Introduction: It's incorrect to state that the Achilles tendon lacks (has no) blood supply or is „bloodless”. It has a poor blood supply.

  11. Introduction: The expressions: „steroid hormone-blocking therapy” and „steroid and lidocaine-blocking therapies” are ambiguous. Please change to „steroid injections”.

  12. Data collection: What does „Achilles tendon stopping point infection” mean? Please, rewrite.

  13. Data collection: „Range of motion” should be written instead of „Movement rates”.

  14. Figure 1 – Part of the text in one of the text windows is lacking.

  15. Theoretically, bias should not occur but should be minimised during the research process.” - This sentence appears twice in a row.

  16. In Meta-analysis, the bias that has the greatest impact on results and is more difficult to control is publication bias, so controlling and identifying publication bias is an important step in Meta-analysis.” The meaning of this sentence is unclear. Please, rewrite.

  17. (...) the same study was excluded based on authorship, a number of cases and interventions and implementation sites. Duplicate publications of the same study were excluded based on authorship, number of study cases and interventions and sites of implementation.” - This information appears twice in a row.

  18. The first passage from the 3.2 section repeats information from the last passage of the 3.1 section and is redundant.

  19. The whole Material and Methods section is too long and wordy. It can be shortened significantly and made more concise.

  20. Figure 2 – Part of the text in one of the text windows is lacking.

  21. Figure 2 – On what grounds were the 290 articles excluded (eg. They did not meet the inclusion criteria based on the Abstracts)? Please include the information in the chart.

  22. Section 3.3.1.2: The Authors state that the Achilles tendon thickness was higher in the PRP group. Was the tendon thickness really higher or was the difference between the Achilles tendon thickness pre- and post-treatment greater? One of the aims of the Achilles tendonitis treatment is to reduce the thickness of the tendon so if it was higher in the experimental group that would mean worsening, not an improvement. Please, explain.

  23. Section 3.3.1.3: Again, the Authors state that VAS scores were higher in the PRP group. Were they really higher or was the difference between the VAS scores pre- and post-treatment greater? The higher the score on the VAS scale, the greater the pain so if the VAS score was higher in the experimantal group that would mean worsening, not an improvement. Please, explain.

  24. How was patient satisfaction determined in the analysed studies?

  25. Table 1 is pasted twice in the manuscript.

  26. Table 1: What do the months in brackets indicate? It is the duration of symptoms before treatment. Please, explain.

  27. What do you mean by „centrifuge training”?

  28. Three months after PRP, they saw a 54% decline, making it difficult to estimate PRP's late effects.” What does the 54% decline refer to?

  29. There is no reference to gingivitis (gums inflammation) in the article by Abate et al. What article are you referring to?

  30. Van der Plas et al. found that 46 patients (58 with AT) rose from 49.2 to 83.6 after 5 years of eccentric exercise.” What do the numbers 49.2 to 83.6 refer to?

  31. The last paragraph before „Unanswered questions and prospects” appears twice in the text.

  32. MSCs abbreviation is not included in the abbreviation list.

  33. The surname of the first author of the first reference is lacking the initial letters. The DOI link of this reference redirects to another article that can be found under reference entry no. 16.

Author Response

Author's Responce

Author's Reply to the Reviewer comments (Reviewer 1) 

Response: We appreciate the Editor's and Reviewers' constructive remarks on our work, which have strengthened it. We responded to the criticisms shown on the attached pages, and the changed text in the manuscript is highlighted in red. We hope the revised work will be accepted for publication in the Journal of Clinical Medicine. 

  1. The manuscript requires English language proofreading and 

editing. 

Response: thanks for your comment.

Two native speakers read the manuscript again, bringing slide modifications or cancelling directly unnecessary phrases or words.

  1. The manuscript requires analysis by a statistical editor.

Response: we are happy about this comment.

The manuscript has been analyzed again by our statistical editor, who brings a clear and better interpretation to this article.

  1. Please, give a quick description of the VISA-A questionnaire 

and VAS scale. What are the treatment aims (higher VISA

A, lower VAS, lower Achilles thickness after treatment, 

the greater difference between Achilles tendon thickness pre

and pos-treatment)? 

Response: we are glad about this comment.

First, the description of the VISA'A and VAS scale is given in each RCT on AT, so we did not add it to this study.

However, in the main time, we will provide it to you as you request.

"VISA-A serves as a disease-specific questionnaire for measuring Achilles tendinopathy severity. The questionnaire is meant to be self-administered. Uncomplicated and relatively quick for both subjects and healthcare professionals. The final version of the VISA-A questionnaire contained eight questions that covered the three domains of pain (questions 1–3), function (questions 4–6), and activity (questions 7 and 8.) Questions one to seven are scored out of 10, and question 8 carries a maximum of 30. Scores are summed to give a total out of 100. An asymptomatic person would score 100. For question 8, participants must answer only parts A, B, or C. If the participant has pain when undertaking sport, he or she automatically loses at least 10, and possibly 20, points."

"The visual analogue scale (VAS) is a validated, subjective measure of acute and chronic pain. Scores are recorded by making a handwritten mark on a 10-cm line representing a continuum between "no pain" and "worst pain."

Second, What does the treatment Aim?

This study aimed to investigate whether Platelet-rich plasma injections can improve outcomes in patients with Achilles tendinopathy.

*VASA'A score, VAS scale score and tendon thickness were all used to assess the outcome of the patients with AT.

We do not administer any treatment in this study. Nevertheless, we analyzed the included studies' treatment results to conclude.

  1. Please, rewrite the Objective for a clear meaning, e.g., The study aimed to determine the efficacy of PRP in the treatment of Achilles tendonitis.

Response: we are grateful for all your comment. We modified the Objective written in red font in the manuscript.

"This study aimed to investigate whether Platelet-rich plasma injections can improve outcomes in patients with Achilles tendinopathy."

  1. How was the Achilles tendon thickness determined in the 

analyzed studies (ultrasound, MRI)? 

Response: thanks for the comment.

According to the studies, the Achilles tendon thickness was determined in the included study By ultrasound and/or by MRI in some cases, but most of the studies used ultrasound which is cheap.

  1. The first paragraph of the Results section of the Abstract belongs to the Methods section and should be transferred there. 

Response: the first part of the result section was modified as requested and was Written in red font in the manuscript. The abstract need to be correct; thanks for your useful comments

  1. What exactly was taken into consideration in the Achilles tendon thickness analysis? Was, is the difference between the tendon thickness before and after PRP treatment? This is not stated clearly at any point in the text. 

Response: We appreciate your comment.

First, the Achilles tendon thickness is assessed using ultrasound or MRI.

Secondly, the result described in our study shows the specific post-treatment status of Achilles tendon thickness measured using ultrasound or MRI according to the included studies. 

 Yes, we did not state it in the text, but the detail about the ATT was inserted directly in figure 3, and now we mentioned that the observation was at 12 weeks of treatment.

  1. The description of the analysis of the VAS score results is unclear. Please, rewrite for a clear meaning both in the Abstract and in the Results section of the manuscript's main body. How would you explain the lack of significant difference in VAS score results after 6 weeks and 6 months but the presence of significant difference after 3 months? Could it be 

due to bias? These results do not justify the statement 

contained in the Conclusions section of the Abstract that 

PRP significantly improved VAS scores. 

Response: we appreciate your comments.

First and most, the non-improvement of the result about the VAS score of the PRP group at 6 and 24 weeks can just be hypothesized to the fact that tendon healing time is longer and that PRP positively affects AT just between 12 to 24 weeks, after which the PRP effect decrease, This is just a hypothesis. The frequency of injection, time of injection, gender and age of injections can all play a role in these results. More randomized control clinical trials are needed for deep understanding and to make a better conclusion.

This is not a Bias, our expert in statistics went through the studies, and this is the result as it appears.

As requested, we modified the Abstract slightly: First result section, then the conclusion, written in red font in the text.

" The VAS improvement results showed no significant difference at 6 and 24 weeks between the 2 groups, respectively (MD = 6.75, 95% CI = (−6.12, 19.62), I 2 = 69%, P = .30), and(MD = 10.46, 95% CI = (−2.44 to 23.37), I 2 = 69%, P =.11). But at 12 weeks of treatment the PRP injection group showed a substantial VAS improvement compared to the control group. (MD = 11.30, 95% CI =( 7.33 to 15.27), I 2 = 0%, P < .00001). The difference was statistically significant. "

The conclusion was rewritten, too, to reduce redundancy.

"There is no proof that PRP injections can enhance patient-functional and clinical outcomes for Achilles tendinopathy. Augmenting the frequency of PRP injections may boost the outcomes, and additionally, More rigorous designs and standardized clinical randomized controlled trials are needed to produce more reliable and accurate results."

  1. Comments to the author: Introduction: "periorbital tissue" is incorrect in this context as it refers to the eye. Please, correct. 

Response: We appreciate your helpful comments. We corrected the text and added a minor modification to the wrong word written in red. 

"Long-term inappropriate or excessive activity causes the Achilles tendon to rub or overstretch beyond its healing capacity and induces inflammatory changes in the tendon and surrounding tissues. ''

  1. Introduction: It's incorrect to state that the Achilles tendon lacks (has no) blood supply or is "bloodless". It has a poor blood supply. 

Response: Response: As you mentioned, we made a mistake because we wanted to reduce the plagiarism rate.

"The Achilles tendon receives insufficient blood supply."

  1. Introduction: The expressions: "steroid hormone-blocking 

therapy" and "steroid and lidocaine-blocking therapies" are 

ambiguous. Please change to "steroid injections". 

Response: We apologize for our mistake; we modified the text as you suggested.

Non-surgical treatment options include steroid injections, oral NSAIDs, external drug application and low-frequency ultrasound stimulation. Steroid injections have strong anti-inflammatory and analgesic effects and are the most widely used treatments in clinical practice.

  1. Data collection: What does "Achilles tendon stopping point 

infection" mean? Please, rewrite.

Response: thanks for your wonderful remarque; as you request, we have modified it.

"(2) Subjects: Patients diagnosed with Achilles tendinopathy".

  1. Data collection: "Range of motion" should be written instead 

of "Movement rates".

Response: we appreciate your impressive comment.

"Movement rates" was removed from the end of the paragraph as it makes no sense.

  1. Figure 1 – Part of the text in one of the text windows is 

lacking. 

Response: thanks for your good remarque.

Figure 1 was removed because it was just supplement material to this article.

  1. "Theoretically, bias should not occur but should be minimized 

during the research process." - This sentence appears twice 

in a row

Response: thanks for your great observation.

we successfully suppressed the repeated sentences.

  1. "In Meta-analysis, the bias that has the greatest impact on 

results and is more difficult to control is publication bias, so 

controlling and identifying publication bias is an important 

step in Meta-analysis." The meaning of this sentence is 

unclear. Please, rewrite. 

Response: We greatly appreciate your helpful comments. We have addressed the Reviewer's comments as indicated below, and the revised sentences are in red font in the manuscript.

"In a meta-analysis, publication bias has the greatest impact on results and is difficult to control. Therefore, controlling and identifying publication bias has become an important and difficult task in meta-analysis."

  1. "(...) the same study was excluded based on authorship, a number of cases and interventions and implementation sites. Duplicate publications of the same study were excluded based on authorship, number of study cases and interventions and sites of implementation." - This information 

appears twice in a row.

Response: we appreciate your comment.

The duplication was suppressed, and a slight modification is mentioned below in red font in the manuscript.

 "Duplicate publications of the same study were excluded based on authorship, the number of case studies, interventions and sites of implementation."

  1. The first passage from the 3.2 section repeats information 

from the last passage of the 3.1 section and is redundant. 

Response: we are grateful for your helpful remark.

Due to the redundancy it causes in the text, we directly suppress the last phrase in section 3.1, as mentioned below.

"A total of 11 publications were screened, of which 11 were in foreign languages and none in Chinese. Three publications were excluded through further evaluation: 1 had an experimental design with no outcome indicators, 1 was a retrospective case-control study and not a true RCT, and 1 had no primary outcome indicators. A total of 8 papers were included in the final results."

  1. The whole Material and Methods section is too long and wordy. It can be shortened significantly and made more concise. 

Response: thanks for this impressive advice.

"A meta-analysis's Material and Methods section are usually longer because authors need to provide a huge amount of data. As you mentioned above, we had much redundancy in this work. After suppressing them, it also helps us reduce the text simultaneously."

  1. Figure 2 – Part of the text in one of the text windows is lacking. 

Response: thanks for this wonderful remark.

"Figure 2 has been successfully modified." 

  1. Figure 2 – On what grounds were the 290 articles excluded 

(eg. They did not meet the inclusion criteria based on the 

Abstracts)? Please include the information in the chart. 

Response: We appreciate your remark. As mentioned below, we added a few pieces of information below in figure 2. 

"Records excluded(n =290 )

Records excluded(n =290 )

Surgery (n =100 )

No control group(n =55)

 Achilles tendon rupture (n =53 )

systemic condition(n =52)

Major injuries to the ankle(n =10 )

combined therapy(n =20)

  1. Section 3.3.1.2: The Authors state that the Achilles tendon thickness was higher in the PRP group. Was the tendon thickness really higher or was the difference between the Achilles tendon thickness pre- and post-treatment greater? 

One of the aims of the Achilles tendonitis treatment is to reduce the thickness of the tendon so if it was higher in the experimental group that would mean worsening, not an improvement. Please, explain. 

Response: we appreciated your comment, which helped us clarify our work.

In this section, we describe the statistical result of 3 previous studies combined, showing that the experimental group had slightly tinner tendons than the control group at three months follow-up. This result was also demonstrated in the 3 RCTs we included in our study. Yes, it means worsening at three months follow-ups, but the difference was not significant, and more research needs to be conducted on this topic.

This is not a mistake because this has been shown by previous studies on the topic, such as the study conducted by Chun-Jie Liu, 

"http://dx.doi.org/10.1097/MD.0000000000015278"

  1. Section 3.3.1.3: Again, the Authors state that VAS scores were higher in the PRP group. Were they really higher or was the difference between the VAS scores pre- and post treatment greater? The higher the score on the VAS scale, the greater the pain so if the VAS score was higher in the 

experimantal group that would mean worsening, not an improvement. Please, explain. 

Response: thanks for your comments. First, we clarify the results of the VAS score on the includes mentioned below and in red font in the text.

"A total of 93 patients in 3 publications correctly and reasonably described the VAS score change for pain at 6,12, and 24 weeks follow-up using a 0 to 100 grading scale. Our results showed no significant difference in VAS score improvement at 6 and 24 weeks between the PRP injection group and the control group, respectively (MD = 6.75, 95% CI = (−6.12, 19.62), I2 = 69%, P = .30), and(MD = 10.46, 95% CI = (−2.44 to 23.37), I2 = 69%, P =.11). But at 12 weeks of treatment (MD = 11.30, 95% CI =( 7.33 to 15.27), I2 = 0%, P < .00001) we observed a significant improvement in PRP treatment group compared to the placebo group."

*To answer your questions, these results were post-treatment 6,12, and 24 weeks', respectively.

*Yes, according to the data we analysed, there was an improvement at three months post-treatment in the PRP group compared to the placebo group. Does it mean it was worsening; no, it was not worsening; the pain actually was reduced in the PRP group at 12 weeks of

 treatment. 

  1. How was patient satisfaction determined in the analysed 

studies?

Response: We appreciate this brilliant question.

Patient satisfaction was determined by giving each a questionnaire after 6,12, and 24 weeks of follow-up. The patients were asked to tick 1 of 2 boxes with the clinical outcome to indicate whether they were "satisfied" or "not satisfied".

 We did not assess patient satisfaction ourselves but used the data in the included studies and made statistical interpretations according to them.

  1. Table 1 is pasted twice in the manuscript. 

Response: thanks for the helpful remark.

We have removed that duplicated table 1

  1. Table 1: What do the months in brackets indicate? It is the 

duration of symptoms before treatment. Please, explain. 

Response: we are happy about your comments.

"Months in the bracket in Table 1 indicated the follow-up time by month as mentioned in our study."

  1. What do you mean by "centrifuge training"?

Response: we are grateful for your comments.

Sorry, it was just a mistake that we corrected and changed to red fond in the manuscript.

 "eccentric training"

  1. "Three months after PRP, they saw a 54% decline, making it 

difficult to estimate PRP's late effects." What does the 54% 

decline refer to? 

Response: We appreciate this helpful remark of yours.

As we noticed it was a mistake from us, we directly modified the form as mentioned below and in red font in the manuscript.

"However, this study's approach is suspect. Patients were blindfolded, making it difficult to estimate PRP's late effects."

  1. There is no reference to gingivitis (gums inflammation) in the 

article by Abate et al. What article are you referring to? 

Response: Thanks for your great comment; this was our mistake.

The word "gingivitis" has been replaced with "tendinopathy", written in red font in our manuscript.

  1. "Van der Plas et al. found that 46 patients (58 with AT) rose 

from 49.2 to 83.6 after 5 years of eccentric exercise." What 

do the numbers 49.2 to 83.6 refer to? 

Response: we are thankful because you have done wonderful work, which helped us improve our article and correct most of the mistakes made involuntarily. We modified the phrase as written below and in red font in the manuscript.

"Found that in 46 patients (58 AT), the VISA-A score significantly increased from 49.2 at baseline to 83.6 after 5 years (p<0.001) of eccentric exercise."

  1. The last paragraph before "Unanswered questions and 

prospects" appears twice in the text. 

Response: We thank you for your comment.

the duplicate was removed thanks

  1. MSCs abbreviation is not included in the abbreviation list. 

Response:  MSCs have been added to the abbreviation list.

 "MSCs: Mesenchymal stem cell."

  1. The surname of the first author of the first reference is lacking 

the initial letters. The DOI link of this reference redirects to 

another article that can be found under reference entry no. 

  1.  

Response: sorry for the mistake, and thanks for your comment.

We modified it as written below and in red font in the manuscript.

"[1] Schneider, Magdalena, et al. "Rescue plan for Achilles: Therapeutics steering the fate and functions of stem cells in tendon wound healing." Advanced drug delivery reviews 129 (2018): 352

375.https://doi.org/10.1016/j.addr.2017.12.016"

Reviewer 2 Report

Thank you for the opportunity to review your manuscript, “the efficacy of Platelet-rich plasma injection therapy in the Treatment of Patients with Achilles tendinopathy: A Systematic Review and Meta-analysis.”

The abstract needs to be shorter and meet the journal's requirements.

The conclusions do not coincide with the results of the meta-analysis. The only statistically significant difference found is in the VAS at three months, which is an increase, so I understand that it is getting worse. Furthermore, a thickening of the tendon is even observed in favour of the PRP group.

The objective of the study needs to be formulated clearly.

Line 108. The data of the ethics committee and registration number are missing.

Line 113-122 - The wording seems more to justify the methodology than to report what has been done. This type of wording should be revised throughout the document. It should be modified to explain what has been done.

The inclusion and exclusion criteria are worded strangely. 

The exclusion criteria cannot be the non-fulfilment of the inclusion criteria.

Line 274-275 - They are similar phrases?

The diagram needs to be redone; the information is not visible. Also, there is information in the flow chart that does not appear in the text, which makes it difficult to understand.

The discussion should be shorter and discuss the aspects outlined in the objectives.

As I have commented above, the conclusions should be reformulated.

Author Response

The efficacy of Platelet-rich plasma injection therapy in the Treatment of Patients with Achilles tendinopathy: A Systematic Review and Meta-analysis

Response: We appreciate the Editor's and Reviewers' constructive remarks on our work, which have strengthened it. We responded to the criticisms shown on the attached pages, and the changed text in the manuscript is highlighted in red. We hope the revised work will be accepted for publication in the Journal of Clinical Medicine. 

Author's Responses

Author's Reply to the Review Report (Reviewer 2) 

  1. The abstract needs to be shorter and meet the journal's requirements.

Response: thanks for your helpful comment.

As you suggested, we have shortened the Abstract, trying our best to be as much as precise.

  1. The conclusions do not coincide with the results of the meta-

analysis. The only statistically significant difference found is in

the VAS at three months, which is an increase, so I understand

that it is getting worse. Furthermore, a thickening of the tendon

is even observed in favour of the PRP group

Response: we appreciate your wonderful comments.

Yes, this was our mistake, and we are sorry for that and glad you noticed.

The conclusion has been rewritten again to be more precise and reduce redundancy.

The modified content is written in red font in the text as below:

"There is no proof that PRP injections can enhance patient-functional and clinical outcomes for Achilles tendinopathy. Augmenting the frequency of PRP injections may boost the outcomes, and additionally, More rigorous designs and standardized clinical randomized controlled trials are needed to produce more reliable and accurate results."

  1. The objective of the study needs to be formulated clearly.

Response: thanks for this comment.

The objective of this study has been modified and written in red fond in the text as follows:

" This study aimed to investigate whether Platelet-rich plasma injections can improve outcomes in patients with Achilles tendinopathy."

  1. Line 108. The data of the ethics committee and the registration

number is missing

Response: we are glad you mention this one.

First of all, we are sorry to make this mistake. It will be cancelled as this is a meta-analysis, not RCT or experimental research.

  1. Line 113-122 - The wording seems more to justify the

methodology than to report what has been done. This type of

wording should be revised throughout the document. It should be

modified to explain what has been done.

The inclusion and exclusion criteria are worded strangely.

The exclusion criteria cannot be the non-fulfilment of the

inclusion criteria.

Response: we are grateful for your comments.

Both Inclusion and Exclusion criteria were reorganized, and the overall format was modified, and we feel satisfied with the results. The modified part is written in red font in the text as follows.

"2.2.1. inclusion criteria: The studies were included if the following criteria were met (1) Study type: Randomized controlled trials (RCTs) published or made public between January 1, 1966, and December 1, 2022; (2) type of patients: Patients diagnosed with Achilles tendinopathy; (3) types of intervention: The experimental group was treated nonsurgically with local PRP injection; (4) Outcome indicators: including the Victorian Institute of Sports Assessment-Achilles (VISA-A) score, change in Achilles tendon thickness, Visual Analogue Scale (VAS) for pain, patient satisfaction, return to sport, and return to exercise. "

"2.2.2. Literature exclusion criteria: have been excluded : (1) Non-randomized controlled trials, low-quality randomized controlled trials, animal and Invitro experiments; (2)studies with Follow up for less than 6 weeks;(3) studies that do not have a control group;(4) Studies with surgical treatment or PRP in combination with other drugs;(5)studies that did not reveal outcomes."

  1. Line 274-275 - They are similar phrases?

Response: thanks for your comments; the exact text has been removed.

  1. The diagram needs to be redone; the information is not visible. Also, there is information in the flow chart that does not appear in the text, which makes it difficult to understand.

The flow chart is a Prisma model chart used by every meta-analysis study to give more specific details about the study search and data selection process; we included, as you requested, the information about the chart flow in the manuscript at "2.4. Literature selection" written in red font as follows.

"From the databases, 405 articles were selected, and 104 Duplicate records were removed. 301 Records were screened, of which 290 needed to be excluded for different reasons (100 studies included Surgery,55 studies had no control group, 53 studies had Achilles tendon rupture,52 studies had other systemic conditions,10 had Major injuries to the ankle, and 20 studies had combined therapy.) 11 articles were then assessed for eligibility. However, during our second screening, we found 3 articles excluded for other reasons: One article was Experimental Research,1 was a Retrospective case study, and 1 had no primary outcome indicator. In the end, only 8 high-quality articles were selected and included in our study."

  1. The discussion should be shorter and discuss the aspects outlined in the objectives

Response: thanks for your comment.

Two authors have worked to bring Modifications in the discussion part, the unnecessary part was cancelled(1/3), and yes, the discussion is more focused on the objective of this study. The modified part is written in red font in the manuscript as follows.

  1. As I have commented above, the conclusions should be

reformulated.

Response: yes, as you mentioned, the conclusion had to be modified, and the modified part is written in red font in the manuscript as follows.

"Our study shows moderate evidence that RPP injection did not significantly improve the VISA-A scores, patient satisfaction, or return to sports rates, and The VAS improvement results showed no significant difference at 6 and 24 weeks between the 2 groups. Nevertheless, the Achilles tendon thickness was higher in the PRP-treated group than in the placebo group at 12 weeks of treatment; the difference was not statistically significant. On the other hand, at 12 weeks of treatment, the PRP injection group showed a substantial VAS improvement compared to the control group, and the difference was statistically significant. Our findings did not support the utility of PRP injection for nonsurgically treated Achilles tendinopathy. It is evident that our study did not reach a consensus with previous research; therefore, we encourage researchers and orthopaedic physicians to keep an open eye on the topics. More rigorous designs and standardized methods are needed to produce more reliable and accurate results. "

Round 2

Reviewer 1 Report

I find the corrections introduced by the Authors mostly satisfactory. However, some of the corrections listed in the Authors' response have not been introduced in the manuscript and a new inaccuracy emerged:

  1. Introduction: The statement that the Achilles tendon „lacks blood supply” has been corrected in the Discussion but not in the Introduction.

  2. I suggest that a short description of VISA-A scale is included in the text of the manuscript. While the VAS scale is widely known, VISA-A is less popular and the presentation of this scale would make it easier for the readers to understand the implications of the described results (similarly to how the Jadad scale is described in passage 2.5).

  3. In passage 3.2 it is stated that: „Eight studies used saline as a control group, and three used blank controls.”, while Table 1 shows that five studies used saline and three used blank controls.

  4. Passage 3.3.1.2: When it comes to Achilles tendon thickness, it would be more accurate to compare the difference in tendon thickness in the study group and control group at the beginning and at the end of the study. If we only compare the thickness at 12 weeks, the conclusion that „the PRP treatment group did not significantly improve Achilles tendon thickness” is not justified because we can not talk about „improvement” when the change in thickness was not assessed over time.

  5. Discussion: „Van der Plas et al. found that 46 patients (58 with AT) rose from 49.2 to 83.6 after 5 years of eccentric exercise”. This correction, although mentioned by the Authors in their response, has not been introduced in the manuscript.

  6. Discussion: The Authors claim that there was an improvement in VAS scores in the PRP-treated group after 12 weeks but in the Discussion, they state: „VAS score of Patients in the PRP group at 12 weeks of treatment was higher than that of the placebo”. A higher VAS score means worsening, not an improvement. This inconsistency needs to be corrected.

  7. The correction regarding Reference 1 has not been introduced.

Author Response

The efficacy of Platelet-rich plasma injection therapy in the Treatment of Patients with Achilles tendinopathy: A Systematic Review and Meta-analysis

Author's Reply to the Reviewer comments (Reviewer 1) 

1. Introduction: The statement that the Achilles tendon "lacks blood supply" has been corrected in the Discussion but not in the Introduction.

Response: Thanks for your comment, and we are sorry for the mistake.

As requested, it was changed and written in red font in the manuscript.

"It heals slower than other injured connective tissues because it has an insufficient blood supply[5]. "

2. I suggest that a short description of VISA-A scale is included in the text of the manuscript. While the VAS scale is widely known, VISA-A is less popular and the presentation of this scale would make it easier for the readers to understand the implications of the described results (similarly to how the Jadad scale is described in passage 2.5).

Response: we appreciate your comment.

As requested, the VISA-A scale was included in the text manuscript and written in red font.

"3.3.1.1. 1. VISA-A score description

A disease-specific questionnaire called VISA-A is used to estimate the severity of Achilles tendinopathy. The survey is designed to be completed independently. For both patients and healthcare professionals, it is simple and rapid. The eight questions in the final VISA-A questionnaire comprised the 3 categories of pain (1-3), function (4-6), and activity (7–8). (7-8.) Questions 1 through 7 are given a score of 10, and question eight can receive up to 30. The sum of the scores yields a score out of 100. Asymptomatic individuals receive a score of 100. Participants must only respond to portions A, B, or C of question 8. A person instantly loses at least 10, and sometimes up to 20, points if they experience pain while participating in a sport."

3. In passage 3.2, it is stated that: "Eight studies used saline as a control group, and three used blank controls.", while Table 1 shows that five studies used saline and three used blank controls

Response: thank you for your wonderful comment; that all help us improve our manuscript.

We are Sorry for that, and the mistake has been corrected.

"Five studies used saline as a control group, and three used blank controls. "

4. Passage 3.3.1.2: When it comes to Achilles tendon thickness, it would be more accurate to compare the difference in tendon thickness in the study group and control group at the beginning and the end of the study. If we only compare the thickness at 12 weeks, the conclusion that "the PRP treatment group did not significantly improve Achilles tendon thickness" is not justified because we can not talk about "improvement" when the change in thickness was not assessed over time.

Response: we thank you for your comment.

Yes, the 3 studies compared the TT at 3 months follow-up, which is inconclusive.

On the other site, we cancelled the part that states that PRP did not "improve", but we state that "the difference was not statistically significant" as the final result shows no statistically significant difference.

5. Discussion: "Van der Plas et al. found that 46 patients (58 with AT) rose from 49.2 to 83.6 after 5 years of eccentric exercise". This correction, although mentioned by the Authors in their response, has not been introduced in the manuscript.

Response: we appreciate your comment and are sorry for the mistake.

The correction was made and written in red font in the manuscript.

"Van der Plas et al. [50]Found that in 46 patients (58 AT), the VISA-A score significantly increased from 49.2 at baseline to 83.6 after 5 years (p<0.001) of eccentric exercise." 

6. Discussion: The Authors claim that there was an improvement in VAS scores in the PRP-treated group after 12 weeks, but in the Discussion, they state: "VAS score of Patients in the PRP group at 12 weeks of treatment was higher than that of the placebo". A higher VAS score means worsening, not an improvement. This inconsistency needs to be corrected.

Response: thanks for your wonderful comment.

We are sorry for the mistake; the PRP group had an improved VAS score; we modified the mistake in red font in the manuscript.

"The VAS score of Patients in the PRP group shows an improvement compare to that of the placebo at 12 weeks of treatment."

7 . The correction regarding Reference 1 has not been introduced.

Response: sorry for the mistake, and thanks for your comment.

We modified it as written below and in red font in the manuscript.

Last time we did not register the modification; sorry for that.

"[1] Schneider, Magdalena, et al. "Rescue plan for Achilles: Therapeutics steering the fate and functions of stem cells in tendon wound healing." Advanced drug delivery reviews 129 (2018): 352

375.https://doi.org/10.1016/j.addr.2017.12.016"

Again we are sorry for all the mistake, and thanks you for being so attentive. We appreciate your contribution to the better of this manuscript.

Reviewer 2 Report

The authors have adequately answered all my questions.

I recommend the authors review the new text, which has typographical errors, such as a lack of capital letters at the beginning of sentences or spaces between words.

I congratulate the authors for their work.

Author Response

The efficacy of Platelet-rich plasma injection therapy in the Treatment of Patients with Achilles tendinopathy: A Systematic Review and Meta-analysis

Author's Reply to the Reviewer comments (Reviewer 2) 

1. I recommend the authors review the new text, which has typographical errors, such as a lack of capital letters at the beginning of sentences or spaces between words.

Thanks for your wonderful work and comments.

Response: Two of our research team members have reviewed the whole manuscript again, and all the little mistakes have been corrected. Thanks for your wonderful contributions to making this a better article; we appreciate it.

Again we are sorry for all the mistake, and thanks you for being so attentive. We appreciate your contribution to the better of this manuscript.